# The Burden of Surgical Site Infection at Hospital Universiti Sains Malaysia and Related Postoperative Outcomes: A Prospective Surveillance Study

**DOI:** 10.3390/antibiotics12020208

**Published:** 2023-01-19

**Authors:** Omaid Hayat Khan, Andee Dzulkarnaen Zakaria, Mohd Nizam Hashim, Amer Hayat Khan, Abdullmoin AlQarni, Manal AlGethamy, Mohammed Mahboob, Anas Mohammed Aljoaid, Nehad Jaser Ahmed, Abdul Haseeb

**Affiliations:** 1Department of Pharmacy, The University of Lahore, Lahore 54590, Pakistan; 2Discipline of Clinical Pharmacy, School of Pharmaceutical Sciences, Universiti Sains Malaysia, Penang 11800, Malaysia; 3Department of Surgery, School of Medical Sciences, Universiti Sains Malaysia, Health Campus, Kota Bharu 16150, Malaysia; 4Infectious Diseases Department, Alnoor Specialist Hospital Makkah, Makkah 24242, Saudi Arabia; 5Department of Infection Prevention & Control Program, Alnoor Specialist Hospital Makkah, Makkah 24242, Saudi Arabia; 6Department of Clinical Pharmacy, College of Pharmacy, Prince Sattam Bin Abdulaziz University, Alkharj 11942, Saudi Arabia; 7Department of Clinical Pharmacy, College of Pharmacy, Umm Al-Qura University, Makkah 21955, Saudi Arabia

**Keywords:** surgical wound infections, surveillance, incidence, predictors, epidemiology

## Abstract

Surgical site infections (SSIs) are the most common healthcare-associated infections that occur among surgical patients. Surgical site infections result in longer hospital stays, hospital readmissions, and higher death and morbidity rates. The current study was designed to highlight the importance of such surveillance studies in a Malaysian surgical population with a motive to evaluate and revise concurrent infection control and prevention policies by exploring the burden of surgical site infection and identifying its associated risk factors for future considerations. In this prospective observational cohort study, a total of 216 patients admitted to a surgical ward were identified and studied. Of these 216 patients, 142 elective procedures and 74 emergency procedures were included in the study, of which 13 patients (9.2%) undergoing elective procedures and 15 (20.3%) patients undergoing emergency procedures were SSI positive (OR: 2.5, *p* = 0.02). Among surgical site infections, 21 were superficial and 7 were deep incisional SSI. No case of organ/space SSI was identified. The time taken for SSIs to develop ranged from 2–17 days with a median of 6 days. Risk factors such as presence of comorbidities (*p* = 0.011), major co-existing medical diagnosis ≥2 (*p* = 0.02), and pre-existing infection (*p* = 0.027) were statistically significant. SSI-positive patients experienced an increase in the post-operative length of hospital stay. In the current population, it was seen that identifying patients who were at high risk of malnutrition via MUST and the NNIS risk index will help clinicians in identifying high risk patients and in managing their patients appropriately. Identifying patients who were at high risk of malnutrition will also improve postoperative outcomes considerably.

## 1. Introduction

Modern-day surgery has been revolutionized completely within the past few decades, but surgical site infection (SSI) still remains an inevitable complication following surgery, even in industrialized countries. In a developing country such as Malaysia, the lack of proper resources, public awareness, adequate infection-control policies, and their implementation remain major setbacks, and thus SSI remains a frequent healthcare-associated infection (HCAI) [1].

SSIs are defined as postoperative infections that occur within 30 days after undergoing a surgical procedure or within 1 year after the placement of a permanent implant [2]. Surgical wounds are classified as clean, clean–contaminated, contaminated, and dirty–infected wounds. Clean infections are uninfected, no inflammation is present, and are primarily closed. Clean–contaminated wounds are wounds that enter the alimentary, respiratory, genital, or urinary tracts under controlled conditions. Contaminated wounds are fresh, open wounds that can result from insult to sterile techniques or leakage from the gastrointestinal tract into the wound. Wounds that typically result from unsuitably treated traumatic wounds are known as dirty–infected wounds [3]. 

Surgical site infections (SSIs) are the most common healthcare-associated infections that occur among surgical patients. According to Geubbels et al., SSIs result in longer hospital stays, hospital readmissions, and higher death and morbidity rates [4]. Badia et al. reported that SSIs raise the risk of death, lengthen hospital stays, increase the readmissions rate, decrease the quality of life, and cause daily income losses, all of which contribute to a family’s financial and social collapse [5].

The World Health Organization estimated that in 2011, the incidence of SSI ranged from 1.2 to 5.2% in high-income nations, while it was 10.8% in low- and low–middle-income countries [6]. According to Chu, the incidence of SSI is 9.4% in high-income nations and 23.2% in low- and low–middle-income countries [7]. In the surgical department of a general hospital in Malaysia, Wong and Holloway reported that the incidence of SSI was 11.7% [8]. According to a study conducted by Fadzwani et al. at the Universiti Kebangsaan Malaysia Medical Centre, the total rate of SSI was 17.18% [9].

The World Health Organization (WHO) considers surveillance as the first and most important step in any infection control and prevention strategy as it helps identify problems and priorities in the system [6]. For this purpose, the current study was designed to highlight the importance of such surveillance studies in the Malaysian surgical population with a motive to evaluate and revise concurrent infection control and prevention policies by exploring the burden of SSI and identifying its associated risk factors for future consideration.

## 2. Materials and Methods

A prospective observational study was conducted among patients admitted to the general surgical wards of Hospital Universiti Sains Malaysia (HUSM) between August 2014 and January 2015. HUSM is the only tertiary care teaching hospital with a bed capacity of more than 800 in Kelantan. It serves an estimated 1.7 million people in its surrounding area in the northeastern part of peninsular Malaysia. The department of surgery has more than four thousand admissions every year, which makes it the third top-ranked department with the highest rate of admissions at HUSM. 

Patients with surgical implants, who died before 30th-day post-operatively without developing SSI, and who failed to consent/follow-up were excluded. The sample size was calculated using the Cochran equation to yield a representative sample for large populations using the prevalence of 11.2% from a previous study [1,10]. By adding a dropout rate of 25%, the final sample size became 191 patients. A total of 426 patients (aged between 18 and 75 years) visited the surgery department during the study period. Among these patients, 401 underwent surgery (the surgery was postponed for 25 patients). Due to the tertiary nature of the hospital, the majority of the patients came upon referral and never returned for a follow-up, and thus a high dropout rate (185 patients) was observed, and the final sample size was cut down to 216 patients. 

Data were collected using a standardized data collection form. A member of the surveillance team (two surgeons, two clinical pharmacists, one nurse, and a nutritionist) reviewed the peri-operative anesthesia flowchart and medical charts (physician’s and nurse’s notes and orders) of all included surgical patients daily. The data included patient socio-demographics, dates of admission, dates of surgery and discharge, necessary surgery-related information, antibiotic prescribing, and related laboratory investigations. Furthermore, upon admission, each patient was stratified according to the National Nosocomial Infection Surveillance (NNIS) and Study on Efficacy of Nosocomial Infection Control (SENIC) risk indexes prior to surgery [11,12]. Additionally, the Malnutrition Universal Screening Tool (MUST) was used to identify patients at risk of malnutrition [13]. The Malnutrition Universal Screening Tool utilizes body mass index (BMI), percentage of unplanned weight loss, and acute disease affect (no nutritional intake for >5 days) to calculate a risk score. The nutritional risk score is categorized as 0 for low risk, 1 for medium risk, and ≥2 for high risk [14]. 

SSIs were diagnosed according to the definitions given by the Centers for Disease Control and Prevention (CDC) [15]. Surgical wounds were inspected by the attending surgeons along with the clinical pharmacist during a daily routine examination. SSI surveillance methods included in-patient and post-discharge surveillance (PDS) until the 30th postoperative day. The PDS methods included (1) direct examination of the patient’s wound during follow-up visits/re-admissions to either surgery clinics/wards or physician’s office, (2) reviewing medical charts of the patients from the surgery or physician clinics other than HUSM, (3) direct patient surveys by contacting them on the telephone. Patients were counseled about their surgical wounds at discharge and were given appointments at 2–3-week intervals and on the 30th day after the operation. Due to limited resources, routine wound cultures were not obtained for every patient, and thus the final diagnosis of SSI was merely clinical. However, patients who underwent culture sensitivity testing were processed without delay using standard microbiological methods [16]. In addition, the surgical care practices that were carried out in HUSM were compared to CDC guidelines for the prevention of surgical site infection [15]. The evaluation of surgical care practices is shown in the Appendix A.

Data were analyzed using the Statistical Package for the Social Sciences (SPSS Inc., Chicago, IL, USA) version 20. Descriptive statistics were used to summarize the demographics, surgical information, and antibiotic data. To remove the bias of patients with increased length of hospital stay (LOS) due to reasons other than SSIs, a median with range was used for LOS. Mann–Whitney U test was used to find the association between LOS and SSI. Some of the continuous variables were grouped into categories quantified as dummy variables such as mean intra-operative body temperature (≤36 °C and >36 °C), percentage of weight loss (<5%, 5–10%, and >10%), duration of surgery (≤120 min and >120 min), LOS prior to surgery (≤7 and >7 days), and LOS post-surgery (≤5 and >5 days) to better understand the distribution of the outcome variables.

In addition to univariate logistic regression (UVLR) analysis, a multivariable logistic regression (MVLR) model was created using the stepwise backward elimination method to predict the independent predictors of SSI. Variables with a significant *p*-value (<0.05) in the univariate analysis and those that were clinically hypothesized to be influential risk factors of SSI, regardless of their statistical significance, were included in the model. Variables such as age and LOS were entered as continuous variables to increase the sensitivity of the multivariate analysis, and those with high collinearity (variance inflation factor > 2) were excluded. A significant association in the MVLR was defined as a *p*-value of less than 0.05.

## 3. Results

Out of a total of 426 procedures performed at the Hospital Universiti Sains Malaysia (HUSM) in two general surgery wards during the study period, only 216 procedures fulfilled the inclusion criteria. The patient recruitment process and categorization into surgical categories/types/specialty are shown in Figure 1. 

The majority of the cases that were lost to follow-up were emergency cases (n = 151) as compared to elective procedures (n = 34). Eventually, 142 elective procedures and 74 emergency procedures were included in the study out of which 13 patients (9.2%) in elective and 15 (20.3%) in emergency procedures were SSI positive (OR: 2.5, *p* = 0.02), rendering an overall incidence rate of ~13% (28/216). A summary of the socio-demographic and surgical characteristics of the study population can be found in Table 1 and in the Appendix A, respectively.

While appendectomy was the most frequently performed procedure (17.1%), followed by wound surgery (Table 2), the majority of SSIs were identified in neurosurgical procedures (17.2%), followed by gastrointestinal surgeries (16.2%) (Figure 2). 

Among the SSIs, 21 were superficial and 7 were deep incisional SSIs. No case of organ/space SSI was identified. About 35.7% of the SSI cases (n = 10) developed after discharge. Among these cases, 6 were diagnosed at re-admission and 4 were identified through other PDS methods. The time taken for SSIs to develop ranged from 2 to 17 days, with a median of 6 days. SSI-positive patients had a median LOS of 13 days as compared to those without SSI, who had a median LOS of 4 days (*p* < 0.001) (Table 3).

### 3.1. Predictors of SSI and the Independent Risk Factors of SSI

The univariate logistic regression revealed that several risk factors such as the presence of comorbidities (*p* = 0.011), co-existing infection (*p* = 0.027), hypoalbuminemia (*p* < 0.001), MUST high-risk group (*p* < 0.001), NNIS risk group (*p* < 0.001), preoperative LOS > 7 days (*p* = 0.009), emergency procedures (*p* = 0.02), surgery duration > 120 min (*p* = 0.021), intraoperative serum glucose level ≥ 10 mmol/L (*p* = 0.008), and postoperative LOS (*p* < 0.001) were statistically significant.

The multivariate linear regression revealed that co-existing infection, MUST high-risk group, NNIS risk group, postoperative LOS, and surgery duration > 120 min were the independent predictors of SSI in general surgery wards at HUSM (Table 4).

### 3.2. Antibiotic Utilization Data

The prescribers followed the recommendation of the antibiotic use guidelines in 86.9% of patients (among the 160 patients, 139 patients received the appropriate antibiotics). The most commonly prescribed antibiotics were third-generation cephalosporins, mainly cefoperazone and ceftriaxone; few patients were administered other classes of antibiotics such as aminoglycosides and carbapenems. The guidelines recommended the use of cephalosporin antibiotics for the majority of surgeries (Table 5). 

A total of 30.6% of patients were given antibiotics for more than 24 h after surgery in contrast to the antibiotic guidelines provided by the Ministry of Health, Malaysia. Among these patients, 85.7% of them received antibiotics for therapeutic purposes but in the remaining 14.3% (n = 7) of patients, antibiotic prophylaxis was prolonged for unknown reasons. Prolonged prophylaxis continued for 2–3 days.

One hundred forty-one patients (88.1%) received surgical prophylaxis within 60 min before the first incision. About 12.8% of the patients who received antibiotics at the appropriate time (n = 18) developed SSIs compared to 21.1% of those who received delayed prophylaxis (*p* = 0.302). The overall SSI rate was 13.75% (Table 6).

### 3.3. SSI-Related Outcomes

SSIs were treated in 21 patients (75%), among which 52.4% were given one drug regimen, a combination of two drugs was given in 33.3% of cases, and more than two drugs were prescribed in 14.3% of patients. Fifteen patients (6.9% of the total population and 53.6% of the total SSIs) received extra or additional antibiotics solely for the purpose of treating SSIs. SSI-positive patients experienced a fivefold increase in the post-operative LOS as compared to non-SSI patients.

Microbiological data were scarce due to financial constraints; however, three patients subsequently underwent culture sensitivity, and isolates revealed the growth of four distinct microorganisms in 3 different patients; Staphylococcus aureus in one, Escherichia coli and Klebsiella pneumonia in another, and multidrug-resistant Acinetobacter in the third. Three mortalities in SSI patients were observed, but only one death was attributable to SSI (mortality rate = 3.6%).

## 4. Discussion

Like any other developing country, Malaysia has a high incidence of SSI, especially when it is compared to industrialized countries [17,18,19,20,21]. According to the WHO, the pooled incidence of SSIs in low- and middle-income countries is 11.8% [6]. The present study revealed an overall incidence of 13%, which is remarkably similar to those previously reported in Malaysia [22]. Such high rates reflect the country’s potential need for improvement in the field as HCAIs, which mirrors the efficiency and robustness of any healthcare system and is considered to be an indicator of the quality and safety being provided to patients [6].

The present study showed that SSIs increase the patient’s length of hospitalization. Misha et al. reported that surgical site infection is linked to a longer hospital stay and a higher number of patients who need to have surgery again [23]. According to Totty et al., SSI was linked to a 92% rise in the duration of hospital stay (*p* = 0.001) [24]. Fenny et al. stated that SSI patients report longer hospital stays than usual for all surgical operations. The longer stays range from 1 day for limb amputation to 16 days for rectal surgery. They came to the conclusion that patients with SSI had considerable hospital stays that lasted longer and increased healthcare costs [25]. Zimlichman et al. and Ban stated that SSI is the most costly type of healthcare-associated infection, with an estimated yearly cost of 3.3 billion USD. It also lengthens the hospital stays of the patients by an average of 9.7 days [26,27]. Moreover, Kirkland et al. and de Lissovoy et al. reported that patients who experience SSI have a doubled risk of death and spend, on average, 10 more days in the hospital [28,29]. According to Whitehouse et al., each orthopedic SSI in their study led to a median increase in hospital stay of 1 extra day during the first hospitalization (*p* = 0.001) and 14 additional days during the follow-up period (*p* = 0.0001) [30].

The rate of neurosurgical SSIs in the present study is strikingly high and stands in contrast to the study previously conducted in Hospital Kuala Lumpur (HKL), which exhibited a considerably lower rate of SSI (7.7%) [31]. Possible explanations for such a contrasting result could be: (1) stringent inclusion criteria in the aforementioned study eliminating the majority of high-risk patients (dirty wounds, pre-existing infections, endoscopic and spinal surgeries, and re-operations), (2) difference of facilities, institutions, surgeons, and surgical protocols, and, most importantly, (3) a very small number of patients (0.3%) had an ASA score of 3 as compared to the present study (31%). Such considerable factors could have potentially underestimated the true incidence at HKL. However, further prospective multicenter studies in both hospitals following a strict surveillance protocol would be able to compare the results of these hospitals more appropriately. According to Haines and Walters, the rate of neurosurgical SSI in any facility should not be higher than 5%, and in our case this rate was exceeded threefold, hence there is an alarming need for major improvements in the present neurosurgical department [32]. On the contrary, no SSI was incident in thyroidectomies, cholecystectomies, splenectomies, and urological surgeries, which can be explained by the use of less invasive techniques such as laparoscopy in the majority of the cases and also by the fact that none of the procedures were contaminated or dirty.

By providing a quantitative analysis of postoperative outcomes, the present study was able to provide evidence-based justifications for considering the number of preventable/modifiable risk factors to be utilized/rectified in current surgical practice. For instance, it is well understood that malnourished patients are at a higher risk of infection, but in the majority of cases, malnutrition remains undiagnosed [33]. Co-existing infection, MUST high-risk group, NNIS risk group, postoperative LOS, and surgery duration > 120 min are the independent predictors of SSI.

In the current population, it was seen that patients at high risk of malnutrition identified through the MUST tool were independently associated with SSI (OR = 3.7), and the situation was similar with patients identified as being at risk by the NNIS risk index (OR = 5.9). MUST and NNIS are tools that are quick, inexpensive, non-labor-intensive, and stratify patients correctly after their admission. Thus, using such tools will help clinicians in identifying high-risk patients and in managing their patients appropriately. Using such tools will also improve postoperative outcomes considerably.

Both the NNIS and SENIC risk indexes have been demonstrated to perform well in the literature [34,35,36]. NNIS was a major improvement over the SENIC project back in 1991; just as the SENIC project replaced the conventional method of wound classification by the National Research Council back in 1985 [11,12,37]. The NNIS risk index also performed well compared to the SENIC (positive predictive value of 24.2 vs. 15.6, respectively) in the present study, which is in concordance with the relative findings of Akin et al. [34]. This result can be attributed to the operation-specific approach of NNIS because instead of a stringent cut-off surgery duration of 120 min in SENIC, NNIS uses a specified cut-off duration for individual surgical procedures that is at the 75th percentile of the type of procedure being carried out. In Malaysia, such risk stratification tools have not yet been practiced or studied, and thus the present study takes the opportunity to utilize such tools and check their reliability in a Malaysian population. Haley highlights the importance of incorporating other procedure-specific risk factors into such indexes in his study in order to increase their performance and reliability [38]. Furthermore, Culver et al. reported that there is still potential for the improvement of such technologies [11]. Thus, it is logical to include other predictors of SSIs, as identified in the present study, in such surveillance indexes to improve their overall performance in a population-specific setting such as Malaysians.

The present study concludes that patients staying more than 7 days in a hospital preoperatively experience approximately a five times greater risk of developing a SSI. Post-operative LOS was an even stronger predictor of SSI because it was independently associated with SSI. In surgical practice, postoperative LOS is of major concern due to an increased chance of attaining an infected wound in a hospital environment. Previous studies have established an association between increased SSI risk and increased pre- and post-operative LOS [17,37,39]. In addition, the financial impact of SSI on the hospital LOS is also not trivial, as infections are the major source of readmissions and a measure of a hospital’s performance [40]. Our data found a five times longer post-operative LOS in SSI patients, which is quite considerable while considering the financial burden, hospital workload, and patient’s quality of life in a developing country.

Overall, surgical care practices at HUSM are quite satisfactory; however, further improvements can be brought about by practicing standard operating procedures as recommended by the international guidelines and by creating awareness regarding the reporting of SSIs and their outcomes, for example by developing officially funded surveillance programs, educating both healthcare professionals and patients, and collaborating with clinical pharmacists. For instance, significant results can be observed if patients undergo antiseptic showering or nasal decolonization before surgery, but such practices are not being practiced at HUSM, and, similarly, hair removal is being carried out using razors instead of clippers or depilatory creams, thus putting patients at risk [41,42,43]. Furthermore, the recommended OR temperature is between 20 and 22.7 °C, but in the present case it was 19 °C [15]. Surgical prophylaxis practices in the current population were quite satisfactory. For instance, 69.4% of patients received antibiotics for the appropriate duration, which is better than the pooled rate of 40.7% in US hospitals [44]. Similarly, according to a previous study in Malaysia and Greece, the rate of antibiotics being prolonged for unknown reasons was as high as 60% and 19%, respectively, but in the present study it was only 14.3% [22,45].

Several studies have reported that the inappropriate use of antibiotics is one of the most serious global public health threats [46,47,48,49,50]. The present study showed that most of the patients received the appropriate antibiotics, and most received surgical antimicrobial prophylaxis within 60 min before the first incision. On the other hand, a total of 30.6% of patients were given antibiotics for more than 24 h after surgery in contrast to antibiotic guidelines. Ng et al. reported that the difference in the overall compliance towards surgical antimicrobial prophylaxis guidelines ranges from 0% to 71.9%. They also reported that the misuse of prophylactic antibiotics is frequently seen and that the main problems are the inappropriate selection of antibiotics and the prolonged duration of administration [51]. According to Alahmadi et al., healthcare providers do not always follow the recommendations for surgical antibiotic prophylaxis. Additionally, they discovered that 56.4% of patients took antibiotics for longer than 24 h [52]. Furthermore, Ahmed et al. found that only 1.08% of the patients in their study received antibiotics adequately for a maximum of one day; the majority of the patients received antibiotics for seven or five days. When compared to the surgical prophylaxis guideline suggestions made by the Saudi Ministry of Health, their analysis demonstrated that there was a significant problem with choosing the right antibiotic and the length of its administration [53].

Regarding the use of antibiotics as surgical prophylaxis, Abdel-Aziz reported that antibiotic prescribing did not match the suggested hospital guidelines in 53.5% of cases. They also reported that prolonged antibiotic use was the commonest reason for non-adherence (59.3%) followed by the use of an alternative antibiotic to that recommended in the protocol (31.5%) [54]. Gouvêa et al. conducted a review on the adherence to guidelines for surgical antibiotic prophylaxis and found that the rate of correct indication of antibiotic prophylaxis ranged from 70.3% to 95%, administration of antibiotics at the appropriate time ranged from 12.73% to 100%, incorrect indication ranged from 2.3% to 100%, adequate discontinuation of antibiotic ranged from 5.8% to 91.4%, appropriate antibiotic selection ranged from 22% to 95%, and adequate antibiotic prophylaxis ranged from 0.3% to 84.5%. They indicated a need for greater adherence to guidelines for surgical antibiotic prophylaxis [55]. Ahmed et al. implemented an antimicrobial stewardship program to improve adherence to perioperative prophylaxis guidelines in Alkharj, a city in Saudi Arabia. They found that after the implementation of their intervention, the appropriateness of drug selection increased from 51.23% to 53.05%, the appropriateness of drug dose improved from 32.72% to 53.66%, the appropriateness of timing increased from 64.81% to 74.39%, the appropriateness of route of drug administration increased from 66.67% to 76.83%, and the duration of prophylaxis increased from 14.20% to 19.51% [56].

Like most observational studies, a few limitations are inherent in the current study and need to be addressed. First, the incidence of SSI in our study may have been underestimated because of an increased number of patients who were lost to follow-up and hence were excluded. Secondly, due to limited resources, we believe the absence of both microbiological testing and institutional policy for antibiotics may have influenced the overall results of this study. Additionally, the incidence calculated in our study may not be meaningful for a few individual surgical categories. The role of shaving the hair in place of hair clipping and the use of alcohol-based abdominal preparation was not analyzed as a determinant of SSI.

Larger, comprehensive, and multicenter surveillance studies including all participating surgical departments are required to help establish an extensive national surveillance database, which will eventually help design prevention and treatment guidelines for SSIs in Malaysia; the present study stresses upon such projects, which may require time, labor, funding, and collaboration between different hospitals but are a dire need of the era.

## 5. Conclusions

SSI-positive patients experienced a fivefold increase in post-operative LOS compared to non-SSI patients. In the current population, it was seen that identifying patients who were at high risk of malnutrition via MUST and the NNIS risk index will help clinicians in identifying high-risk patients and in managing their patients appropriately. Identifying patients who were at high risk of malnutrition will also improve postoperative outcomes considerably.

## Figures and Tables

**Figure 1 antibiotics-12-00208-f001:**
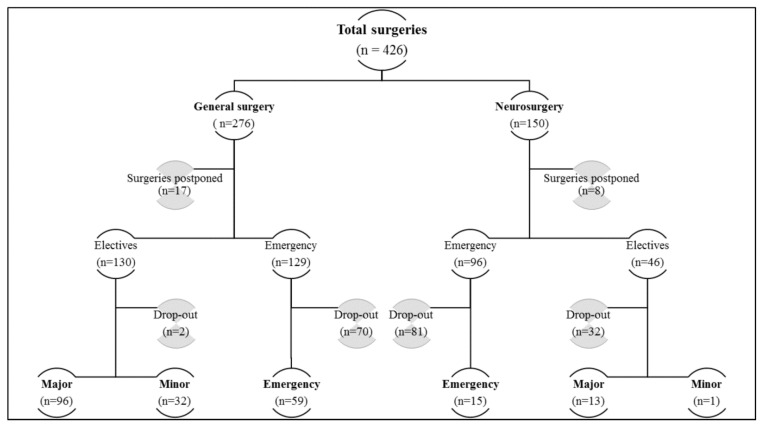
Detailed flowchart of patient recruitment throughout the study period displaying the drop-out of the patients who were lost to follow-up and hence were excluded.

**Figure 2 antibiotics-12-00208-f002:**
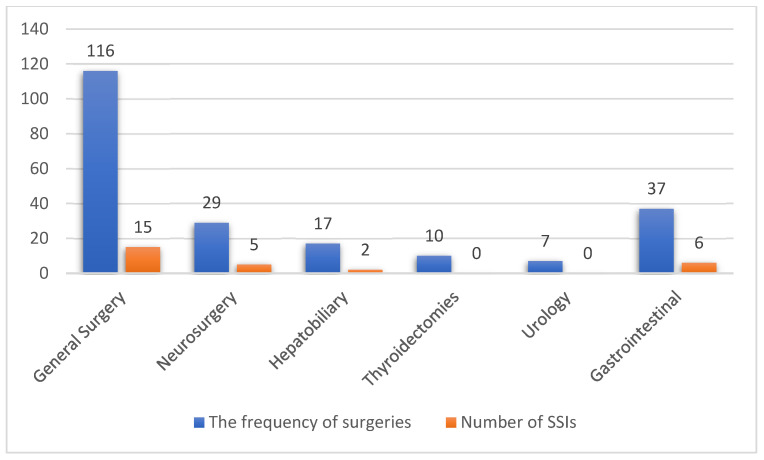
The incidence of SSIs. The bar chart represents the number of surgeries by category and the number of surgical site infections (SSIs) for each category.

**Table 1 antibiotics-12-00208-t001:** Socio-demographic characteristics of the study population (n = 216).

Characteristics	Number (%)
Age (years)	48 (17–75) *
Gender	
Male	113 (52.3)
Female	103 (47.7)
BMI (kg/m^2^)	
≥30 (Obese)	36 (16.7)
25.0–29.9	51 (23.6)
18.5–24.9	97 (44.9)
<18.5	32 (14.8)
Race	
Malay	199 (92.1)
Chinese	14 (6.5)
Others	3 (1.4)
Smoker	
No	150 (69.4)
Yes	66 (30.6)
Drinks Alcohol	
No	213 (98.6)
Yes	3 (1.4)

* Median with interquartile range; BMI, body mass index.

**Table 2 antibiotics-12-00208-t002:** The numbers and percentages of surgeries.

Surgical Category	The Numbers and Percentages of Surgeries N (%)
General surgery	116 (53.7)
Appendectomy	37 (17.1)
Wound surgeries *	36 (16.7)
Herniorrhaphy	21 (9.7)
Exploratory laparotomy	5 (2.3)
Splenectomy	3 (1.4)
Mastectomy	14 (6.5)
Neurosurgery ^†^	29 (13.4)
Hepatobiliary	17 (7.8)
Cholecystectomy	15 (6.9)
Biliary surgeries	2 (0.9)
Thyroidectomies	10 (4.6)
Urology ^‡^	7 (3.2)
Gastrointestinal	37 (17.1)
Colectomies	19 (8.8)
Gastrectomy	6 (2.8)
Others ^§^	12 (5.6)
Total	N = 216

* Includes 19 incisions and drainage of abscesses, 6 debridements of assaulted wounds, 4 excisions of lipomas, 2 saucerizations, and 4 secondary wound closures; ^†^ Includes 28 craniotomies and 1 laminectomy; ^‡^ Includes 3 removal of stones (ROS) and 4 transurethral resection of prostate (TURP) procedures. ^§^ Other gastrointestinal surgeries include haemorrhoidectomy, small bowel surgery, oesophagectomy, and rectal resection surgeries.

**Table 3 antibiotics-12-00208-t003:** Impact of SSIs on patients’ length of hospitalization (LOS).

Description	Distribution between Cohorts Median (Range)		*p*-Value ^†^
	SSI	Non-SSI	
Pre-operative length of stay (days)	1 (1–17)	1 (1–16)	0.107
Post-operative length of stay (days)	10 (1–79)	2 (1–79)	<0.001
Total length of stay (days)	13 (2–80)	4 (1–95)	<0.001

† Mann–Whitney U test.

**Table 4 antibiotics-12-00208-t004:** Multivariate analysis of the data reveals independent predictors of SSI in the current surgical population.

Variables	N	Surgical Site Infections (%)		Univariate *p*-Value	Multivariate *p*-Value	Adjusted Odds Ratio	95% Confidence Interval	
		Yes (n = 28)	No (n = 188)				Upper	Lower
Age	216	49 ± 14.6 *	44.7 ± 16.8 *	0.201	0.886	0.997	0.961	1.035
Gender (female)	103	39.3	48.9	0.342	0.965	0.978	0.361	2.646
Comorbidity	104	71.4	44.7	0.011	0.121	2.571	0.780	8.480
Co-existing infection	111	71.4	48.4	0.027	0.036	3.127	1.080	9.050
Emergency procedures	74	53.6	31.4	0.024	-	-	-	-
Hypoalbuminemia	54	57.1	20.2	0.001	-	-	-	-
Intraoperative serum glucose ≥ 10 mmol/L	17	21.4	5.9	0.008	-	-	-	-
Operations by junior surgeon	96	42.9	44.7	0.856	0.059	3.228	0.955	10.912
High risk of malnutrition (using MUST)	46	50	17	0.000	0.012	3.764	1.339	10.582
Patients at risk with NNIS	99	85.7	39.3	0.000	0.003	5.975	1.813	19.698
Pre-operative LOS (days)	216	3.9 ± 5.0 *	1.9 ± 2.0 *	0.002	-	-	-	-
Post-operative LOS (days)	216	15.9 ± 18 *	5.23 ± 8.44 *	0.000	0.038	1.040	1.002	1.079
Surgery duration > 120 min	80	57.1	34	0.007	0.027	3.571	1.156	11.035

Goodness of fit = 0.905 (Hosmer and Lemeshow test). * Mean with standard deviation.

**Table 5 antibiotics-12-00208-t005:** The recommended prophylactic antibiotics for different surgical categories.

Surgical Category	The Recommended Antibiotics
Obstetrics and gynecology surgeries Head and neck surgeries Plastic surgeries Vascular surgeries Gastrointestinal surgeries Orthopedic surgeries Urological surgeries Neurological surgeries Cardiac surgeries Hepatobiliary surgeries	Cephalosporin alone or with metronidazole Cephalosporin alone or with metronidazole Cephalosporin aloneAmpicillin/sulbactam or amoxi cillin/clavulanic acid Cephalosporin alone or with metronidazole Cephalosporin alone Cephalosporin alone or with metronidazole Ampicillin/sulbactam, or amoxicillin/clavulanic acid Cephalosporin alone or with metronidazole Cephalosporin alone Cephalosporin alone Cephalosporin and gentamicin

**Table 6 antibiotics-12-00208-t006:** The overall SSI rate.

Item	Number of the Patients	SSIs Number	Rate of SSIs
Patients who received antibiotics at the appropriate time	141	18	12.8%
Patients who received antibiotics at an inappropriate time	19	4	21.1%
Total	160	22	13.75%

## Data Availability

Not applicable.

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
