# Peer review of "The Burden of Surgical Site Infection at Hospital Universiti Sains Malaysia and Related Postoperative Outcomes: A Prospective Surveillance Study"

_antibiotics, 2023, doi:10.3390/antibiotics12020208_

Round 1

Reviewer 1 Report

Introduction need to be elaborated with definition and types and prevalence of SSI .Material and methods should be prior to the results and after introduction in the manuscript.Role of Shaving of the hair in place of hair clipping and use of alcohol based abdominal preparation has not been analysed as a determinant of SSI. International recommending bodies has shown the significant role of these two with body shower prior to the surgery as a significant measures to prevent SSI.  

Author Response

Introduction need to be elaborated with definition and types and prevalence of SSI .

We add paragraphs about the definition and types and the prevalence of SSI

Material and methods should be prior to the results and after introduction in the manuscript.

we change the order of the study sections as recommended 

Role of Shaving of the hair in place of hair clipping and use of alcohol based abdominal preparation has not been analysed as a determinant of SSI. International recommending bodies has shown the significant role of these two with body shower prior to the surgery as a significant measures to prevent SSI.

we add this as limitations  

Reviewer 2 Report

The article only need to be revised by native English speaker

Author Response

The article only needs to be revised by native English speaker

we rewrite the paper

Reviewer 3 Report

very interesting article.

Author Response

we make the recommended modifications

Reviewer 4 Report

Thank-you for this interesting paper. Please see comments below.

Abstract#22 and 23 -opening sentences have no relation to introduction. this study does not focus on c/section which is what the absract suggests. Please revise opening sentence

Introduction: There is no review of the literarture about SSI in Malaysia or comparable countries. Indeed we are unaware of the prevalence of SSI in OECD countries. This must be included to provide better context for this study.

Results

#56 - expand HUSM 

#57 what does 2 Intan and 3 Utara mean?

Figure 1 needs to show the final numbers in the SSI and non SSI group. This means lines #64-69 can be heavility edited ot removed.

Table 1 needs to be moved to methods

What does Indians etc mean? This is not specific

Smoking habits should read Smoker?

Drinking Habits should read Drinks Alcohol?

Table 2 should be a supplementary table

Duration of surgfery should all be in minutes

There is no criteria provided for junior or senior surgeons (e.g > 5 years)

Table 3 needs to become a figure with 2 parts

a) a graph showing incidence of SSI and then a table showing surgical category, details of the surgery (the long notes sectrion will become redundant) anf the frequency of the surgery. 

There are far too many tables and the results are confusing,.

Section 2.1 has no relevance to the results. Consider creating a supplementary methids section. 

Section 2.2 ishoudl be tabulated or create an infogram - its impossible to read and there is important imformation here

 Section 2.4 -rational and irrational prescribing - this is not clear. I assume from reading this means following the government guidlines or not following them. Please rephrase.

Also this would be a key component of the paper and of revelance to readers - this section lacks detail. In fact including the culture plating and resistance would be good in this section  - see below re: 2.5

Secttion 2.5 should be incorporated into 2.4 . The mortality is addressed in the next section.

The discussion needs to be clearer - there is information there which needs to be in the introduction and needs to focus on the key findings

- increased LOS for SSI

- prerdictors of SSI

- Ab utilisation

- how does it compare

- what to do next

Minor grammar changes/queries

#44 As for a developing should be In a

#45 - awareness - is this public awareness - this needs to be expanded

#46 - "thus SSI is still a" should read "thus SSI remains a"

#64 emergent should read emergency

#78 It was seen appendectomy should read While appendectomy was the most frequently performed procedure, followed by wound surgery, the majority of SSIs were identified...

Author Response

Abstract#22 and 23 -opening sentences have no relation to introduction. this study does not focus on c/section which is what the absract suggests. Please revise opening sentence

we rewrite the abstract part

Introduction: There is no review of the literarture about SSI in Malaysia or comparable countries. Indeed we are unaware of the prevalence of SSI in OECD countries. This must be included to provide better context for this study.

we add paragraphs about the definition of SSI and its prevalence 

Results

#56 - expand HUSM we expand it

#57 what does 2 Intan and 3 Utara mean? we delete the words it is in Malay language

Figure 1 needs to show the final numbers in the SSI and non SSI group. This means lines #64-69 can be heavility edited ot removed.

we edit the required lines

What does Indians etc mean? This is not specific we rewrite it

Smoking habits should read Smoker?  we rewrite it

Drinking Habits should read Drinks Alcohol? we rewrite it

Table 2 should be a supplementary table  we move it to supplementary

Duration of surgfery should all be in minutes we rewrite it

There is no criteria provided for junior or senior surgeons (e.g > 5 years) we we write the criteria

Table 3 needs to become a figure with 2 parts

a) a graph showing incidence of SSI and then a table showing surgical category, details of the surgery (the long notes sectrion will become redundant) anf the frequency of the surgery. 

we make the required modification

There are far too many tables and the results are confusing,. we change a table to figure and we move one of the table to the supplementary

Section 2.1 has no relevance to the results. Consider creating a supplementary methids section. 

we add it as a supplementary 

Section 2.2 ishoudl be tabulated or create an infogram - its impossible to read and there is important imformation here

we modified it

 Section 2.4 -rational and irrational prescribing - this is not clear. I assume from reading this means following the government guidlines or not following them. Please rephrase.

we rewrite it

Also this would be a key component of the paper and of revelance to readers - this section lacks detail. In fact including the culture plating and resistance would be good in this section  - see below re: 2.5

we modified itit

Secttion 2.5 should be incorporated into 2.4 . The mortality is addressed in the next section.

we rewrite it

The discussion needs to be clearer - there is information there which needs to be in the introduction and needs to focus on the key findings

- increased LOS for SSI

- prerdictors of SSI

- Ab utilisation

- how does it compare

- what to do next

we rewrite it and add limitations 

Minor grammar changes/queries  

we made the required grammar changes 

Round 2

Reviewer 1 Report

replace word decrease with improve in the abstract line 38 of pdf file.

Author Response

We replaced it

Reviewer 4 Report

The authors have addressed efficiently the comments, however, I have some minor changes to tables and figures.

Line 150 - 51 Figure 1 elaborates on the patient’s recruitment process and surgical procedure subdivisions should read patient recruitment process and categroisation into surgical categories/types/speciality are shown in Figure 1.

The legend for Figure 2 needs to be detailed. The legend should be read in its own right i.e all acronymns should be spelled out eg bar chart of number of surgeries by category and surgical site infections (SSI for each category.

In Table 3 please unbold preoperative length of stay (days) and please remove total median range column from the table as this column does not provide any additional information. 

Author Response

Line 150 - 51 Figure 1 elaborates on the patient’s recruitment process and surgical procedure subdivisions should read patient recruitment process and categroisation into surgical categories/types/speciality are shown in Figure 1.

We modified it

The legend for Figure 2 needs to be detailed. The legend should be read in its own right i.e all acronyms should be spelled out eg bar chart of number of surgeries by category and surgical site infections (SSI for each category.

We modified it

In Table 3 please unbold preoperative length of stay (days) and please remove total median range column from the table as this column does not provide any additional information. 

We unbolded preoperative length of stay and we removed total median range column